# Sustainable Tourism Development and Ramsar Sites in Serbia: Exploring Residents' Attitudes and Water Quality Assessment in the Vlasina Protected Area

Ana Milanović Pešić *[ID], Tamara Jojić Glavonjić, Stefan Denda [ID] and Dejana Jakovljević

Geographical Institute "Jovan Cvijić", Serbian Academy of Sciences and Arts (SASA), 11000 Belgrade, Serbia; t.jojic@gi.sanu.ac.rs (T.J.G.); s.denda@gi.sanu.ac.rs (S.D.); d.jakovljevic@gi.sanu.ac.rs (D.J.)
* Correspondence: a.milanovic@gi.sanu.ac.rs

**Abstract:** This study aims to present the potential for sustainable tourism development on Vlasina Lake, which is, along with its surroundings, declared as a Ramsar site, Natural Asset of Exceptional Importance, IBA, IPA, PBA and Emerald area. A survey conducted among the residents indicated that they expressed positive attitudes towards sustainable tourism development, even though a small percentage of them are employed in tourism. Considering the lake as the most valuable part of this area, this study emphasized water quality assessment as the necessary condition for sustainable tourism development. Water quality indices (SWQI, CWQI and WPI) were used for water quality assessment for the period 2013–2022. Based on SWQI, Vlasina Lake has a good to excellent water quality and, according to WPI, has clean water suitable for tourism and recreation. The CWQI for overall water quality ranged from marginal to good. It is the highest for recreation, but it is based on only one parameter (pH), which is the limitation of this methodology. Based on field research, survey, water quality assessment and previous studies, it is concluded that this area has favorable conditions for developing various types of tourism, which could contribute to the future development of this undeveloped and unpopulated area.

**Keywords:** Ramsar site; residents' attitudes; water quality; sustainable tourism; Vlasina Lake; Serbia

## 1. Introduction

Freshwater ecosystems, including lakes and wetlands, are considered essential life assets. However, these ecosystems occupy relatively limited parts of the Earth [1]. These aquatic ecosystems have diverse socio-economic and ecological functions besides providing a habitat to aquatic life [1]. According to the World Lake Vision [2], lakes are an essential component of global water resources. The lake's role is fundamental in the continuing cycle of evaporation, precipitation and water flow on and under the land surface. Lakes are storage bodies for large water quantities, sources of food and recreational pleasure for humans, as well as habitats for various aquatic organisms. During flood events, lakes have the ability to mitigate flood waves and, in that way, protect lives and properties. Lakes also have aesthetic value and represent some of the most beautiful features of the landscape [2]. However, wetlands and lakes are fragile ecosystems with a sensitive functional ability [1]. Anthropogenic pressures in the catchments of these ecosystems have resulted in water quality deterioration and biodiversity threat [1,3,4].

Various studies addressed many impacts on lakes: nutrient pollution from agriculture and wastewater, plastic pollution, industrial waste discharges, climate change effects, invasive species and habitat destruction [5–7], hydrological alteration [8], tourism activities [9], fishing [10], illegal mining activities [6], urbanization [11], and population growth [12,13], acidification [14] and eutrophication [15]. Although the aquatic ecosystems are limited in quantity and very sensitive to external impacts, most are mismanaged and neglected natural resources. These water bodies require proper management and care [1]. The

international network Living Lakes aims to improve the protection, restoration and rehabilitation of lakes, wetlands and other freshwater bodies and their catchment areas [16]. Their objectives are to conserve the biodiversity of lakes, lake regions and wetlands, as well as the valuable ecosystems provided by lakes and wetlands; to preserve saline water and freshwater resources, including lake and wetland ecosystems; to restore degraded and disappearing wetland and lake ecosystems, to improve the life quality of local communities, sustainable use and development of wetland ecosystems; to promote applied sciences and technologies for the conservation of these ecosystems; to support educational programs and cooperation with local communities towards the conservation of these ecosystems and biodiversity; and to disseminate information and to increase awareness regarding the multiple values of these ecosystems [16].

The reservoir that is the focus of this paper—the Vlasina Lake—was built for electricity production and water supply. This lake is, along with its surroundings, under protection as a Natural Asset of Exceptional Importance, and it was declared as a Ramsar site. In addition to criteria such as the presence of settlements in the protected area, the representation of the traditional method of natural resources use and the protection status for more than ten years, the Vlasina Natural Asset of Exceptional Importance (Vlasina LEP) was selected as the subject of this study because of its location. It is situated in one of the most underdeveloped and depopulated parts of Serbia. Also, it is a border area, which has established cooperation on natural resource use with neighboring Bulgaria, which could lead to the development of municipalities in both countries. As a member of the European Union, Bulgaria has access to many development funds, including those regarding environmental protection. Also, Serbia has access to some of these funds, especially the IPA fund. In order to use EU funds, cooperation between the two countries began through IPA programs of cross-border cooperation, firstly for road infrastructure renewal and then through projects related to natural and cultural heritage, ecology and environmental protection. As an outcome of a joint project "Green border area–investment in nature", of the municipality of Surdulica in Serbia and the municipality of Ćustendil in Bulgaria, the Information Center for Environmental Protection was built in Vlasina Rid and opened in May 2015. In addition, another valuable project from the tourism aspect was successfully implemented in the period 2016–2018. This project was entitled "Development of tourism in the border region of Bulgaria and Serbia by creating tourist attractions and exhibition of representative cultural and historical sites of the municipalities Surdulica and Pravets" [17,18]. All of the abovementioned indicates that the Vlasina LEP has a regional character and represents the potential for the economic development of the wider area in both countries. Therefore, studies like this can be useful to decision-makers at both the national and international levels.

Although Vlasina Lake is the second largest in Serbia, to the authors' knowledge, it has not been the subject of detailed water quality analysis using statistical water quality index methods. In this paper, for the first time, a water quality assessment of Vlasina Lake was carried out by applying the Canadian Water Quality Index (CWQI) and Water Pollution Index (WPI), which enabled us to determine the water quality in this protected area, in terms of sustainable tourism development, as well as to point out the pollution dangers. In addition, lake tourism is not particularly developed in Serbia, and there is a lack of studies dealing with this type of tourism. Therefore, studies that cover lake tourism are important for its development in Serbia. Also, it is important to emphasize that this study aims to analyze residents' attitudes about their involvement in tourism activities and perspectives for future sustainable tourism development in the Vlasina LEF. Therefore, the results of this study will serve scientists in the adequate further protection of Vlasina Lake and the preservation of good water quality, as well as serving decision-makers about the prospects for economic development in this economically undeveloped and depopulated area.

Considering the abovementioned, Vlasina Lake is a key resource for tourism development in this area. All tourist activities are connected directly to the lake (sports-recreational, fishing tourism) or for its neighboring surroundings (ecological tourism, hunting tourism,

excursion tourism). Therefore, it is very important to analyze the water quality of this lake because, without it, there would be no conditions for tourism development in this area. On the other hand, the residents represent a key factor that can enable tourism development, and it is important to include their attitudes toward this issue. Therefore, the main research issue in this study refers to establishing a connection between water quality and residents' attitudes according to the development of tourism as a potential basic activity.

In order to study these subjects, the following research hypotheses were proposed in this paper:

**Hypothesis 1 (H1).** *A small number of residents are involved in the tourism business.*

**Hypothesis 2 (H2).** *The socio-demographic characteristics of residents have an impact on residents' attitudes toward the sustainable tourism development.*

**Hypothesis 3 (H3).** *Vlasina Lake is a reservoir exposed to anthropogenic pressures and erosion processes, and water quality deteriorates over time.*

**Hypothesis 4 (H4).** *Tourism has significant development potential in the future period.*

## 2. Literature Review

Taking into account the multifunctional role of lakes and wetlands, as well as their sensitivity to various pressures, many lakes and wetlands worldwide are under the protection of the Ramsar Convention. Due to their ecological, biological, educational, social and economic importance, Ramsar sites have always attracted significant attention from researchers. Studies of various Ramsar sites have been conducted worldwide. Sustainable tourism development has significantly contributed to the valorization of these areas.

Sustainable tourism management strategies are possible to develop through understanding residents attitudes, needs and rights [19,20]. Surveys of residents' attitudes towards the tourism contribute in long-term planning and management growth, protecting community values and developing nature-based, cultural and historical attractions [21]. Cultural tourism plays an important role in economic and sociocultural sustainability by helping the local community to increase employment and income, to preserve cultural and historical monuments, strengthen the local identity and improve everyday life [22]. On the other hand, unplanned and poorly managed tourism can damage natural environment, causing environmental pollution and pressure on local infrastructure, habitats and resources [23].

Residents have a critical role in the tourism planning and development process. Residents' perceptions of tourism development are considered as a vital step for participatory tourism plans in the Greek part of Prespa Lake. The results show weak engagement of residents in participatory opportunities but also their willingness to be actively included in the decision-making process [24]. Sustainable management of Ramsar sites has been studied in Songor Lagoon Ramsar site in Ghana [25]. Residents were mainly not aware of the economic benefit of Ramsar sites for the community and they have identified the following main factors of degradation: waste disposal, poor attitudes of residents toward environmental protection, wildfires, shoreline recession, small-scale industries, fishing, farming and lack of the public education about impacts of environmental degradation on Ramsar sites. Perceptions of ecotourism potential from residents have been studied in the Lake Natron Ramsar site; the results show that Lake Natron has the potential for ecotourism development but lacks a general management plan, mechanism for the fair distribution of ecotourism benefits, developed tourist infrastructure facilities and adequate funding [26]. Analyses of wetland as sustainable tourism destinations have been conducted in the Kilombero Valley Ramsar site in Tanzania; the results show that landless people have more negative attitudes toward wetland tourism than landowners [27]. The perceptions and attitudes of residents toward wetland conservation have been analyzed in the Xuan Thuy National Park Ramsar site in Vietnam; the residents have positive attitudes towards

wetlands conservation and a willingness to participate in conservation activities; however, their awareness of the threatments of wetlands is not high [28]. Threats and public attitudes towards nature conservation have been also identified in Bumdeling in Bhutan. The results show positive attitudes towards wetland conservation and their willingness to contribute and inform people about the importance of conservation. The major threats are agricultural activities, overgrazing, lack of budget and human resources [29]. The residents of Mare Aux Cochons in the Seychelles perceived ecotourism, biodiversity conservation and environmental protection as the most important activities and their positive attitudes towards biodiversity conservation are caused by the frequent visits and benefits from ecosystem services [30]. Residents' perceptions and preferences towards natural resources have been studied in U Minh Thoung National Park in Vietnam; the results show that habitat is the most highly valued benefit, while the ecotours services provided limited resident participation, so recreation and tourism are underrated [31]. Residents' perceptions of the impacts of drought on wetland and household benefits have been studied in the Driefontein Grasslands in Zimbabwe; the findings show that wetlands ecosystem services and wetland-based agriculture activities are severely affected by frequent droughts which occur at least once in two years [32]. Residents' willingness to pay for restoration has been analyzed in Ashtamudi Lake in India, and the findings show that they have a willingness to pay for modest and moderate wetland improvement scenarios. The results also indicated that the highest value for residents was in mangrove conservation, followed by water quality and sustainable fishing [33]. A study of anthropogenic impacts on Ramsar sites has been performed in the Koshi Tappu Wetland in Nepal. The findings show that the following activities harm the wetland ecosystem: the use of inorganic pesticides and fertilizers in agriculture, picnic activities, wildlife observation, firing, fishing and hunting [34].

In Serbia, eleven wetlands are on the List of Internationally Important Wetlands (Ramsar sites) [35]. All of them are under national protection and classified in category I—natural assets of exceptional importance. The connection between tourism and nature protection has been explored in various Ramsar sites in Serbia: Gornje Podunavlje, Slano Kopovo, Zasavica, Labudovo Okno. The results show that wetlands with all natural phenomena including canals, lakes, ponds and the habitats of rare and endangered animal and plant species are the main potential for ecotourism development, while the disrupted water regime and embankment constructions are potentially dangerous for these Ramsar sites [36]. Gornje Podunavlje and Koviljsko-Petrovaradinski rit have been the subject of study of residents' attitudes and perceptions towards nature conservation. The results indicate positive attitudes towards protection, but residents feel excluded from the protected area management [37]. The socio-economic potential of the Ramsar site has been examined in the Carska Bara Special Nature Reserve. The results indicated an unfavorable demographic structure, with a predominantly old population; regardless of the natural potential for tourism, there is no implementation of significant measures for the improvement of existing conditions [38]. This Ramsar site was also the subject of examination of the residents' perceptions toward the protected area and showed a low level of understanding and dialogue between residents and the managing body [39]. The landscape of exceptional features "Vlasina" has also been the subject of the study of residents' perceptions toward the protected area and the findings confirmed that the designation of the protected area has little or no impact on their life and economy [40]. Vlasina Lake has been studied from the aspects of the impacts of tourism on the lake water quality during 2008, 2010, 2011 and 2020. The results show that the water was very clean, with some exceptions during 2020. Also, the study confirmed that the lake should be protected from the pollution from septic tanks by the construction of a sewage system [41].

Socio-economic and cultural factors have important roles in sustainable water resources management. Water-related tourism and recreation may be affected by water quality deterioration [42]. In order to assess lake water quality, water quality indices methodology is applied worldwide. The Serbian Water Quality Index (SWQI) was applied in the following studies in Serbia: the lakes Ludoš and Palić [43], and the artificial lakes

(reservoirs): Bela Crkva, Bačka Topola, Moravica, Zvornik, Grlište, Bor, Bojnik, Vlasina, Ćelije, Ovčar Banja, Međuvršje, Bovan, Krajkovac, Pridvorica, Zavoj, Bresnica, Divčibare, Bajina Bašta, Kokin Brod, Vrutci, Sjenica, Potpeć, Radoinja, Zlatibor, Garaši, Grošnica, Prvonek [43], Gruža [43,44] and Barje [43,45]. Some studies about the Danube River quality involved artificial Đerdap Lake [43,46–49]. The Canadian Water Quality Index (CWQI) is applied in various studies of lakes and artificial lakes (reservoirs): the Polyphytos artificial lake in Greece [50], Wadi El-Rayan lakes in Egypt [51], Qu' Appelle Valley lakes in Canada [52], Lake Kinneret in Israel [4], reservoirs and lakes in the Ebinur Lake Watershed in China [53], accumulation lakes on the Olt River and Danube River in Romania [54], Djerdap Lake on Danube River in Serbia [46,49], and the Palić and Ludaš Lakes in Serbia [55,56]. The Water Pollution Index (WPI) is applied for water quality assessment in Fuxian Lake in the protected area in southwest China [57]; in Poyang Lake, the largest freshwater lake in China [58]; Ulansuhai Lake, the largest shallow lake of the Yellow River in China [59]; Qiandao Lake in China [60]; Soyang reservoir in South Korea [61]; Chungju Reservoir in South Korea [62]; the lakes Burabai and Ulken Shabakty in Kazakhstan [63]; Nasser Lake in Egypt [64]; the lake in the near od Dramaha Campus of IPB in Indonesia [65]; Lower Vaengskoye Lake in the Murmansk Region in Russia [66]; and Djerdap Lake on the Danube River in Serbia [49,67,68].

Considering that lakes often have recreational roles and possible pollution can affect human health, various lakes have been studied for their appropriateness for recreational purposes: Shahu Lake in China: the results showed that workers (on and around the lake) were more exposed to carcinogenic and non-carcinogenic risks than tourists, but the risks were in acceptable limits and thresholds and the lake water quality was safe for tourists and workers in 2013 [69]; the artificial lake Kisköre Reservoir (Lake Tisza) in Hungary: the results showed the good ecological status of this lake which enabled an increased number of tourists leading to increased anthropogenic loads [70]; Lake Ostrovąs in Poland: this lake is classified as hypertrophic with a bad ecological status and should not be used for recreational purposes [71]; Lake Ełk in Poland: the findings showed that the water quality in 2015 and 2016 was slightly improved (reducing the nutrients and decreasing algal blooms and improving visibility) comparing with the period from 1999 to 2008 [72]. Natural conditions, based on water temperature measurements (from 1961 to 2020) for the development of lake tourism have been analyzed in Poland. The results show an earlier beginning and later ending of bathing seasons in recent years, which could affect the water quality [73]. The impacts of tourism activities on water pollution have been investigated in the West Lake Basin in China. The results showed that water pollution increased (from 2007 to 2018) due to an increased number of tourists which led to increased garbage, and that the government, due to economic benefits, neglected the pollution generated from tourist activities [74].

Lake tourism has been the subject of various studies around the world. It has also been presented at lake tourism conferences, which were held in: Savonlinna in Finland (2003), Thousand Islands Lakes in Hangzhou in China (2005), Gyöngyös in Hungary (2007), and Thunder Bay in Canada (2009) [75]. Tourist perceptions have been analyzed in the tourist area of Lake Toba in Indonesia. The results show that the development of tourism quality was underway in 2020, but a significant change in the perceptions of foreign tourists has been notable, compared with 2004 [76]. Residents' and tourist perceptions have been studied in Lake Mjøsa in Norway. The results suggest the following sustainable experience dimensions: interactions with the natural and cultural environment, insights and views and lake-based activities [77]. Tourism development and sustainability have been studied in Lake Salda and its environment in Turkey. The results show that Lake Salda has recreational capacity for lake tourism due to landscape, water quality and suitable environmental structure, but it lacks organization, promotion and research [78].

In Serbia, lakes and reservoirs account for 8% of captured water [79]. Unlike natural lakes that are smaller in size and importance, reservoirs are of great importance and most often have multifunctional purposes. Their purpose is primarily to supply water

to the population and industry, energy, irrigation, flood protection and traffic. They are used to a lesser extent for fishing needs and, more recently, for sustainable tourism development [79–81].

## 3. Study Area

Vlasina Landscape of Exceptional Features (Vlasina LEF) is located in the southeastern part of the Republic of Serbia, near the border with Bulgaria and Vlasina Lake (Figure 1). It covers eight settlements of the Surdulica municipality and a small part of the Crna Trava municipality (without settlements), with a total area of 13,329.84 ha [82]. This area is 10 km away from the international Corridor X (Belgrade-Niš-Skopje-Thessaloniki), and it is crossed by several regional directions (M 1.13, R 122, R 124a, R 124b) [83].

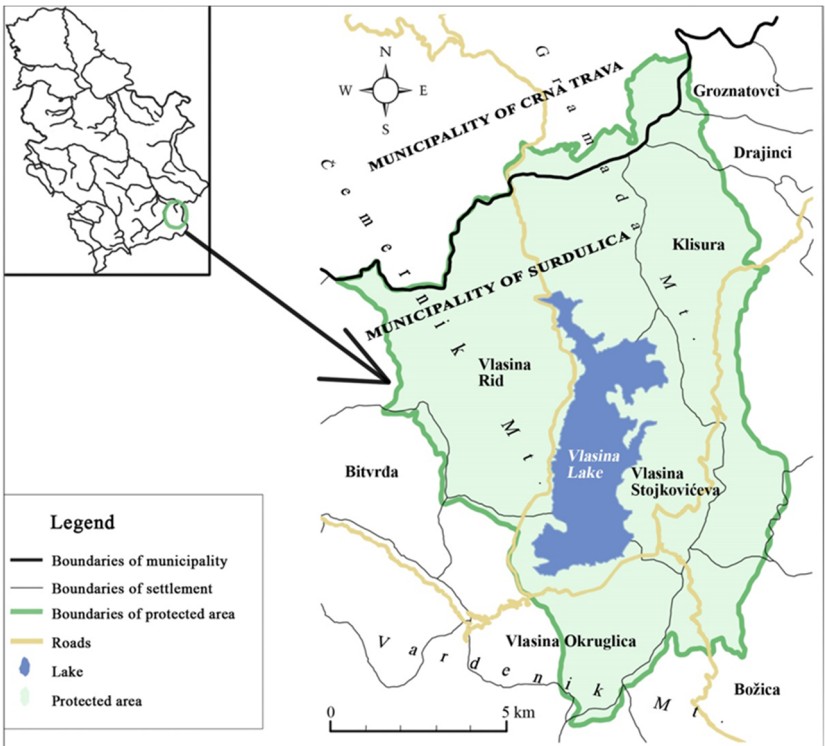

**Figure 1.** Map of the study area.

The main attraction of the protected area is Vlasina Lake (Figure 2). It is an artificial lake, the highest lake in the Republic of Serbia (1213.8 m a.s.l.) and the second largest (16 km²) [84]. A reservoir was formed in 1949 [85]. It is 9 km long, 1.8 km average width (maximum 3.5 km) with a depth of up to 35 m and an indented shore of 132.5 km length [86]. It is surrounded by the mountain ranges of Čemernik (Veliki Čemernik, 1638 m), Vardenik (Veliki Strešer, 1876 m), and Gramada (Vrtop, 1721 m) [87]. The unusual feature of the lake are two islands—Stratorija and Dugi Del—the so-called peat islands that represent a unique curiosity. These are temporary islands, i.e., torn-off parts of the sedge that float up from the bottom of the lake or are torn off from the mouth of the tributaries of Vlasina Lake. They represent a natural rarity, as habitats of a specific living world, and therefore are accessible only to researchers with special permission.

On a national level, Vlasina Lake and its 500 m-wide surrounding area was protected for the first time in 1999 by a decision on prior protection. In 2006, due to diverse biodiversity and natural resources quality, this lake was, along with a greater part of the Vlasina plateau, the surrounding settlements of the Surdulica municipality and a smaller part of the Crna Trava municipality, declared a Natural Asset of Exceptional Importance and protected as a Protected Area of Category I [88]. There have been recorded 840 species of plants,

about 140 species of birds, 80 species of vertebrates, and 28 species of mammals in this area [89,90].

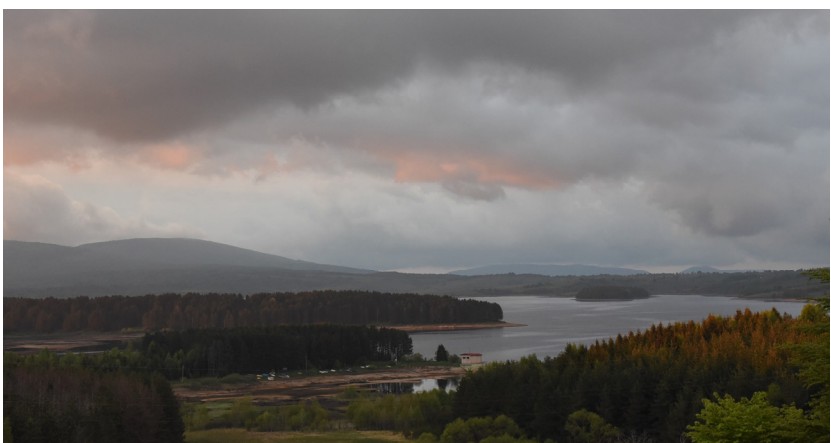

**Figure 2.** Vlasina Lake.

In international frameworks, this area is protected as an Important Plant Area (IPA), an Important Bird Area (IBA), the Prime Butterfly Area (PBA), and part of the European ecological network (Emerald). According to the IUCN categorization, it belongs to category V—Protected terrestrial/marine area. It has a particular value as a Ramsar site, and it is one of three Ramsar sites located south of the Sava and Danube, in the central part of Serbia, while the other nine are located in the Autonomous Province of Vojvodina (northern part of Serbia).

The Vlasina LEF is also rich in cultural heritage. Among many, there are the Palja monastery (10th–11th century), the former medieval monastery on Rid, and today the church of the Holy Prophet Elijah, as well as the remains of medieval surface mines, the so-called "dumps", which testify to the developed mining in Roman times [91].

In terms of outdoor recreation, the climatic features in the Vlasina LEF are not the most favorable for all activities and not throughout the year. Due to the altitude, winters are very cold, springs are colder and rainier than autumn, and summers are cool [92]. Another characteristic of this area is the high windiness. Nevertheless, in the warmer period of the year, the landscape characteristics of Vlasina LEF offer excellent potential for tourism development. Exceptional landscapes with their orographic, hydrological, and biogeographical characteristics are suitable for mountaineers, mountain runners, cyclists, paragliders, orienteering competitors, swimmers, kayakers, hunters, fishermen and collectors of medicinal herbs and forest fruits. Along with the folk heritage of this region, they form the basis for holding numerous events. Although artificial, the lake is perfectly integrated into the environment [84,91]. All of the abovementioned natural and cultural heritage values create conditions for the use of this area for scientific-educational, cultural and tourist-recreational purposes [82], which is especially significant because this is a depopulated and economically underdeveloped area of Serbia [93].

## 4. Data and Methodology

### 4.1. Residents' Attitudes

The first part of this research includes the analysis of a survey conducted among residents in the Vlasina LEF. The survey was part of broader research on the experience of everyday life in a protected area. Models for the questionnaire were found in previously published foreign and domestic studies dealing with residents' attitudes to protected areas. Trakolis [94] examined the awareness of residents of the protected area, personal economic income thanks to the protected area and the number of tourists and reasons for their attendance after declaring the protection of the area. In his study, Nolte [95] analyzed how establishing and managing the protected area affected residents' daily lives,

focusing on the negative aspects. Christopoulou and Tsachalidis [96] explored residents' attitudes about the management and exploitation of the wetlands, with a focus on their values and tourism and agriculture development. Also, Alkan et al. [97] assessed the general attitudes of the residents about protected areas, primary factors causing positive or negative local perceptions of protected areas and the involvement of the local community in decision-making in protected areas. Sladonja et al. [98] examined the conservation knowledge of residents and their perception of protected areas, leadership activities and management authorities. In Serbia, Tomićević et al. [99] examined residents' attitudes about the quality of life after the declaration of a protected area and the economic perspective for living in that area in the future, and Pavić et al. [100] assessed residents' awareness about the protected area, their attitudes toward protection and tourists, as well as their suggestions for the sustainable development of the protected areas. Based on a review of this literature [94–100], a survey instrument was developed for this part of the study. Also, the questionnaire was developed in consultation with reports about protected areas and by contacting researchers in institutions focused on protected areas' management.

The original questionnaire contained 32 questions, and 13 were separated and analyzed for this study to examine residents' attitudes toward tourism activities in this area and its future development, as well as their engagement in tourism. The part of the survey set aside for this particular study contains mostly closed-ended questions with fixed responses, and several yes/no questions and multiple-choice questions. The survey obtained categorical data that were analyzed with the Statistical Package for the Social Sciences, version 20, using descriptive statistics and chi-square ($X^2$) analysis. This involves the use of descriptive demographic statistics, as well as the chi-square test of independence, in order to compare the collected data in terms of gender, age and education.

A survey was conducted in 2017 on a sample of 81 randomly selected respondents from all settlements of the Vlasina LEF (Božica, Vlasina Okruglica, Vlasina Rid, Vlasina Stojkovićeva, Groznatovci, Drajinci, Klisura). Although it seems that the sample is small, it is important to emphasize that it represents about 9.6% of the adult population in this territory. According to the 2011 Census, 945 residents lived in LEF settlements, of which 845 were adults [101]. Questionnaires have been distributed with the help of the three local elementary schools. Both sexes are equally represented among the respondents, and the average age of all adults in this sample is 47. According to the socio-economic structure, most respondents are employed (Table 1). The survey was conducted on a voluntary basis of respondents, completely anonymously, with no personal data (e.g., health, ethnicity, political opinion, religious or philosophical conviction), and all participants were adults. The interviewees were informed who was conducting the research, in which institution and that the obtained results would be used exclusively for scientific research.

**Table 1.** Socio-demographic characteristics of the respondents.

| Age | | |
|---|---|---|
| | Mean | 46.96 |
| | Minimum | 18 |
| | Maximum | 73 |
| **Gender** | **Frequency** | **Valid percent** |
| Male | 41 | 50.6 |
| Female | 40 | 49.4 |
| Total | 81 | 100.0 |
| **Employment status** | **Frequency** | **Valid percent** |
| Employed | 48 | 59.3 |
| Unemployed | 26 | 32.1 |
| Retired | 7 | 8.6 |
| Total | 81 | 100.0 |

Source: Authors' research.

In the meantime, from 2017 to the present day, the infrastructure and organization in this protected area have remained the same. Only the number of inhabitants has decreased even more. According to the 2022 Census, 578 inhabitants live in these settlements, of which 536 are adults [102]. Their average age increased. The protected area's boundaries were expanded, including one more settlement (Bitvrđa), but the number of inhabitants did not increase (the added settlement has only five inhabitants). Based on all of the abovementioned, it can be concluded that the data obtained in the survey are still valid and current.

### 4.2. Water Quality

In order to determine whether the water quality of Vlasina Lake is suitable for sustainable tourism development, an assessment of the water quality was made by using three indices in the second part of the study. Water Quality Index methodology is a suitable tool for this purpose, processing various parameters of water quality, and has been applied in numerous studies around the world. Data on water quality at the Vlasina Lake were obtained from the Institute of Occupational Safety from Novi Sad for the period 2013–2022. Parameters were measured once (2017 and 2019), twice (2016, 2021 and 2022) and three times (2013, 2014, 2015, 2018 and 2020). The small number of samples is one of the limitations in providing detailed results about water quality in this study. The data were processed applying the following water quality indices: the Serbian Water Quality Index (SWQI), Canadian Water Quality Index (CWQI) and Water Pollution Index (WPI). Each of these water quality indices is based on a set of parameters, and some of the parameters differ among different indices. In order to obtain more complete insights into water quality, all of these indices are applied.

The Serbian Water Quality Index (SWQI) was developed by the Serbian Environmental Protection Agency (SEPA) and contains ten quality parameters: oxygen saturation (OS), biochemical oxygen demand (BOD), ammonium ($NH_4$-N), pH, total nitrogen oxides (TNO), orthophosphate ($PO_4^{3-}$), suspended solids (SS), temperature (T), conductivity and coliform bacteria (CB). It is calculated as follows:

$$SWQI = \sum q_i \times w_i \qquad (1)$$

where the $q_i$ is the value of each parameter, while the $w_i$ is the weight unit [103,104].

The calculation for each parameter is based on parameter values. The SWQI value is a dimensionless single number, ranging from 0 to 100, within five categories [103] (Table 2). This index is useful for the assessment of organic and nutrient pollution. However, this index could not be used in inorganic pollution assessment because it does not include heavy metals. The relatively small number of parameters is its main limitation.

The Canadian Water Quality Index (CWQI) was developed by the Canadian Council of Ministers of the Environment, based on the British Columbia Ministry of Environment formulation, in 1995 [105]. This methodology is based on following parameters: color, turbidity (Turb), dissolved oxygen (DO), pH, calcium (Ca), sodium (Na), sulphate ($SO_4^{2-}$), chloride ($Cl^-$), fluoride ($F^-$), nitrate, nitrite ($NO_3^-$, $NO_2^-$), aluminium (Al), arsenic (As), barium (Ba), beryllium (Be), cadmium (Cd), chromium (Cr), Copper (Cu), iron (Fe), mercury (Hg), manganese (Mn), molybdenum (Mo), nickel (Ni), lead (Pb), zinc (Z). It is calculated using the Canadian Water Quality Index 1.0 Calculator (EXCEL application). For each CWQI range, a descriptive quality indicator has been defined. The CWQI is suitable for the assessment of inorganic pollution. In addition to the overall water quality, this index is used to assess water quality for specific purposes: drinking, aquatic life, recreation, irrigation and livestock.

**Table 2.** Water quality assessment according to the SWQI, CWQI and WPI index.

| SWQI | | CWQI | | WPI | |
|---|---|---|---|---|---|
| **Value** | | **Value** | **Description** | **Value** | |
| 90–100 | Excellent | 95–100 | Excellent | $\leq 0.3$ | Very pure |
| 84–89 | Very Good | 80–94 | Good | 0.31–1.0 | Pure |
| 83–72 | Good | 65–79 | Fair | 1.01–2.0 | Moderately polluted |
| 39–71 | Bad | 45–64 | Marginal | 2.01–4.0 | Polluted |
| 0–38 | Very Bad | 0–44 | Poor | 4.01–6.0 | Impure |
| | | | | >6.01 | Heavily impure |

Source of data: [103,105,106].

The Water Pollution Index (WPI) is applied for the assessment of the ecological, chemical and biological water quality of Vlasina Lake. It presents a combined index, including physical, chemical and biological elements for water quality assessment. For analyzing WPI index in this study, the data of 24 physical, chemical and biological parameters were included: dissolved oxygen (DO); oxygen saturation (OS), pH; suspended solids (SS); biochemical oxygen demand (BOD); chemical oxygen demand ($COD_{MnO4}$); total organic carbon (TOC); nitrite ($NO_2^-$); nitrate ($NO_3^-$); ammonium ($NH_4$-N); total nitrogen (TN); chloride ($Cl^-$); sulfate ($SO_4^{2-}$); orthophosphate ($PO_4^{3-}$); total phosphorus (TP); metals (Fe, Mn, Cu, Zn, Cr, B and As) dry residue in the water and coliform bacteria (CB). *WPI* is calculated based on the formula [106]:

$$WPI = \sum_{i=1}^{n} \frac{Ci}{SFQS} \times \frac{1}{n} \tag{2}$$

where *Ci* is the measured annual average value of a parameter; *SFQS* is the prescribed maximum value of the parameter for the I water quality class in Serbia (for rivers and natural lakes) and for the II/III class (for reservoirs), and n is a number of used parameters.

The prescribed threshold values for all parameters for the given classes in Serbia are established at the national level by the Rulebook on the Parameters of the Ecological and Chemical Status of Surface Waters, and the Parameters of Chemical and Quantitative Status of Ground Waters [107], the Regulation on Emission Limit Values for Pollutants in Surface and Ground Waters and Sediments and the Deadlines for Their Reaching [108], and the Regulation on Limit Values of Priority Substances and Priority Hazardous Substances Polluting Surface Waters and the Deadlines for Their Reaching [109].

The WPI is one of the simple indicators for water pollution estimation, providing a good explanation of the main pollution factors in various water bodies. Its advantage is that there are no limitations on the number and types of the used parameters. Therefore, the WPI is widely used to evaluate the water quality status in different water resources, including lakes and reservoirs.

Water quality assessment is based on the calculated values of the SWQI, CWQI and WPI indexes (Table 2).

## 5. Results and Discussion

Lakes are among the most important aquatic ecosystems but are also the most sensitive to anthropogenic pressure. Both natural lakes and reservoirs often represent important tourist destinations or potential resources for the development of different types of tourism. Therefore, the attitudes of the residents or tourists about various aspects of the lake's ecological problems, environmental protection, development and the impact of tourism on them are often analyzed. In this context, the attitudes of residents living in the in Vlasina LEF about the protected area and tourism development are analyzed in this study, and the obtained results could serve as an important indicator of the state of the environment. On the other hand, the anthropogenic impact significantly modifies the quality of nature resources. Vlasina Lake represents the fundamental value of this protected area and a

key resource for tourism development because all tourist activities directly depend on the lake or its neighboring regions. Therefore, it is very important to examine its water quality, because without it there would be no condition for tourism development. In this regard, the implementation of the water quality indexes methodology has a significant role, as it allows us to determine the current water quality, register pollutants and indicate potential sources of pollution. In addition, the assessment of the water quality of Vlasina Lake is an important step towards its tourism development based on the principle of nature protection.

*5.1. Survey of Residents' Attitudes*

5.1.1. Residents' Attitudes toward Tourism Development

In order to examine the engagement of the residents in the tourism sector, economic benefits and their attitudes toward the future development of sustainable tourism in the Vlasina LEF, several distinct impact variables were analyzed. According to the obtained results in Table 3, it can be concluded that a small number of residents are engaged in tourism (19.8%). Of that number, 37.5% started dealing with tourism before the announcement. The largest share of respondents (77.8%) has never been engaged in tourism and is not engaged in it now either. The prevailing opinion is that after receiving the status of a protected area, the number of visitors increased (44.4%). In line with that, 58.3% of respondents advocate that position and consider it a direct consequence of the announcement, while the other 41.7% do not. The largest number of respondents (67.9%) have an opinion that the number of visitors is low, but there are also those (3.7%) who think that there are too many of them. The obtained results suggest that the hypothesis (H1) related to the involvement degree of residents in the tourism industry has been accepted. The main reason for such a situation is the low level of tourism development in the previous period. However, they believe that the future development of this area is directly related to this activity. A significant number (71.6%) believe this would also contribute to reducing emigration from this region. Further, most of them think that dealing exclusively with tourism cannot be a sufficient income for a decent life.

**Table 3.** Residents' attitudes toward tourism activities and its future development.

| Residents' Attitudes | | Frequency | Valid Percent |
|---|---|---|---|
| Are you engaged in tourism? | Yes | 16 | 19.8 |
| | No | 65 | 80.2 |
| I am engaged in tourism: | From getting the status | 6 | 7.4 |
| | Even before the area was protected | 10 | 12.3 |
| | Not now, but I used to | 2 | 2.5 |
| | Never | 63 | 77.8 |
| I believe that engagement in tourism can provide sufficient income for a decent living. | Yes | 16 | 19.8 |
| | Yes, but not only through tourism | 50 | 61.7 |
| | No | 15 | 18.5 |
| I believe that tourists visit Vlasina LEF: | Lower | 23 | 28.4 |
| | Same as before | 22 | 27.2 |
| | Higher, due to status | 21 | 25.9 |
| | Higher, not due to status | 15 | 18.5 |
| I believe that the number of tourists in Vlasina LEF is: | Low | 55 | 67.9 |
| | Sufficient | 23 | 28.4 |
| | Too many | 3 | 3.7 |
| Vlasina's future is in the development of tourism | Yes | 60 | 74.1 |
| | No | 21 | 25.9 |

In addition to examination of the residents' attitudes toward tourist attendance before and after the declaration of Vlasina as a protected asset, it is also interesting to analyze their

views on the impact of the protected area on their everyday lives. The conducted research in this aspect showed that 96.3% of respondents are aware that they live in a protected area, and 58% believe that this has a positive impact on their everyday lives. As negative effects of the protected area declaration on their everyday life, some of the respondents stated that they could no longer fish freely (12.3%), can no longer extract peat from the lake (3.7%) and have problems with flooding (7.4%) [40].

5.1.2. The Impact of the Residents' Socio-Demographic Characteristics on Their Attitudes toward Tourism Development

In order to examine the relationship between the socio-demographic variables and attitudes related to the tourist activity of residents, the value of Cramer's V and phi coefficients have been used to determine the extent of the mutual size effect. The obtained results showed that 49.4% of women and 50.6% of men are engaged in tourism. As the ratio is very similar, the first test showed no statistically significant relationship between the variables of gender and tourism ($p = 0.956$). The attitude that tourism is the only activity which is sufficient for existence is expressed by 22.5% of women and 17.1% of men. The highest percentage of both groups (67.5% of women and 56.1% of men) believe that tourism as the only activity is not enough for existence, while the rest of women (10.0%) and men (26.8%) believe that tourism in Vlasina cannot provide sufficient income. Crossing these two variables in the contingency table showed that the differences by gender are not significant ($p = 0.148$) when it comes to this attitude. In addition, gender is not significant regarding the attitude about the existence of conditions for tourism in the Vlasina LEF ($p = 0.690$). A small percentage of respondents from both categories stated they have the conditions to engage in tourism (f = 20.0%; m = 19.5%). Also, there is an equal number of women and men who neither have the conditions nor want to engage in tourism (40.0% of women and 48.8% of men). The crossing of the gender variable with the attitude of respondents about tourism as a factor that could reduce emigration from the settlement of the Vlasina LEF showed that there is no connection ($p = 0.503$), that is, that the respondents of both genders believe that the development of tourism would contribute to migration decrease (f = 75.0%; m = 68.3%). The opinions of women and men who live in the Vlasina LEF are almost the same regarding the attitude that tourism development in this area can attract new settlers to their settlements (f = 66.7%, m = 68.3%). The absence of differences or connections between the variable gender and the variable tourism as a factor of attracting a new population was confirmed by the chi-square test of independence ($p = 0.877$). The results shown in Table 4 indicate that gender is not statistically significantly related to the attitude that the future of the Vlasina LEF is in tourism development ($p = 0.749$). Namely, 72.5% of female and 75.6% of male respondents believe that tourism will contribute to the development of this region. In contrast, an approximate percentage of respondents of both genders (27.5% and 24.4%) believe the opposite.

Since the age range of the respondents was from 18 to 73, for this study analysis, two categories were distinguished—young people (18–44 years) and older people (45–73 years). The chi-square test did not show a relationship between the population's age and the tourism practice ($p = 0.185$). The results' analysis showed a difference, but it is still not significant. In the youth category, 12.5% of the total number are engaged in tourism, while in the category of older people, 24.5% of the total number are engaged in tourism. The value of the phi coefficient (−0.147) unequivocally confirms this. Crossing the age variable with the variable related to the possibility of a decent existence from tourism activities also showed no connection ($p = 0.139$). Both categories mostly have the opinion that tourism as the only activity cannot bring enough income. This was especially emphasized by younger respondents (75.0% of young people and 53.1% of older people). Crossing the variables of age and condition for tourism activities shows that the conditions that residents have or do not have for engaging in tourism do not depend on their age. This was confirmed by a non-parametric test ($p = 0.402$) and a similar percentage of respondents in both categories who declared that they had the conditions to engage in tourism (12.5% of young people and

24.5% of older people). The percentage of young and old residents who do not have the conditions and do not want to engage in tourism is similar (46.9%), as well as the percentage of those who do not have the conditions but would engage in tourism if they received government subsidies (40.6% and 32.7%). Tourism as a factor in emigration reduction is another variable that does not depend on age. This is undoubtedly confirmed by the data on 78.1% of young people and 67.3% of older people who answered affirmatively on this question and the significance obtained by the chi-square test ($p = 0.293$). The highest percentage of respondents believe that future tourism development could attract people to settle in the Vlasina LEF. According to this attitude, there is almost no difference between the age groups (65.6% of the young and 61.1% of the older people), which is also confirmed by the chi-square test ($p = 0.770$). Both categories mostly believe that the future economic progress of the Vlasina LEF is in tourism development. This attitude is held by a slightly higher percentage of older respondents (79.6%) compared to young people (65.6%), but the chi-square test did not show a connection ($p = 0.161$), and neither did the phi coefficient ($-0.156$). Therefore, the differences in frequency are the result of chance.

**Table 4.** The socio-demographic characteristics of the residents and their attitudes toward tourism.

| Residents' Attitudes | Socio-Demographic Characteristics | Phi Coeff | Cramer's V Coeff. * | $\chi 2$ ** ($p$ Value) |
|---|---|---|---|---|
| Are you engaged in tourism? | gender | 0.006 | 0.006 | 0.956 |
| | age | −0.147 | 0.147 | 0.185 |
| | education | 0.182 | 0.182 | 0.260 |
| I believe that engagement in tourism can provide a decent living | gender | 0.217 | 0.217 | 0.148 |
| | age | 0.221 | 0.221 | 0.139 |
| | education | 0.245 | 0.173 | 0.301 |
| I believe I have the necessary conditions to engage in tourism | gender | 0.096 | 0.096 | 0.690 |
| | age | 0.150 | 0.150 | 0.402 |
| | education | 0.261 | 0.184 | 0.239 |
| I believe that the development of tourism can keep residents in Vlasina LEF | gender | 0.074 | 0.074 | 0.503 |
| | age | 0.117 | 0.117 | 0.293 |
| | education | 0.288 | 0.288 | 0.035 |
| I believe that the development of tourism can attract new residents to settle in Vlasina LEF | gender | −0.017 | 0.017 | 0.877 |
| | age | −0.033 | 0.033 | 0.770 |
| | education | 0.128 | 0.128 | 0.518 |
| I believe that the future of Vlasina LEF is in tourism | gender | −0.035 | 0.035 | 0.749 |
| | age | −0.156 | 0.156 | 0.161 |
| | education | 0.200 | 0.200 | 0.199 |

Note. * In the case of 2 × 2 table, the value of the phi coefficient is interpreted ** Significant: $p \leq 0.05$ (at level 95%).

The difference in the education level of residents and their engagement in tourism was not significant ($p = 0.260$). However, respondents with a high school education (25.0%) led in engagement in the tourism sector compared to those with university education (13.3%) or those with only elementary school (7.1%). The acquired level of education and the attitude towards tourism as an activity that can enable a better life in settlements of the Vlasina LEF are unrelated ($p = 0.301$). In terms of percentages, all three categories of respondents (with primary, secondary and university education) predominantly believe that tourism can only be an additional source of income. Residents with a university education mostly agree with this (86.7% compared to 57.1% of those with elementary school and 55.8% with high school). The level of education is not related to the property, i.e., the lack of conditions for engaging in tourism ($p = 0.239$). However, there is a slightly higher percentage of those with a high school education (25.0%) who stated that they have the conditions to engage in tourism compared to those with an elementary school education (7.1%) and those with a university education (13.3%). However, crossing the collected data on the respondents' education level with their attitude about tourism as a factor in reducing emigration in the Vlasina

LEF showed a statistically significant connection between these two variables ($p = 0.035$). This is also confirmed by the percentage of those who believe in the benefit of tourism development. Almost 93.3% of respondents with a university education have that opinion, while the percentage of respondents with elementary school who share this opinion is significantly lower (50.0%). The value of Cramer's coefficient (0.288) for these variables is very close to a medium size effect. Residents' education does not affect their opinion on whether the tourism development in the Vlasina LEF can attract new residents. This is also shown by the chi-square test ($p = 0.518$) and percentages. A slightly higher percentage of those with a university education (80%) have the attitude that tourism development can attract new residents, compared to those with high school (64.7%) or elementary school (64.3%). Nevertheless, the descriptive statistics clearly show that the highest percentage of those who see the future of the Vlasina LEP in tourism development is among respondents with an elementary school education (92.9%) than with a high school education (71.2%) and with a university education (66.7%). According to the results, it can be noticed that the connection between these variables still exists, as shown by the value of Cramer's coefficient (0.200), which is closer to medium than to weak in strength. Based on all of the above, it can be concluded that Hypothesis H2 was not accepted as was expected. Namely, the opinion that respondents' attitudes towards tourism development depends on their gender and age differences was rejected. Similar results were achieved in relation to the educational characteristics of the respondents. The exception is the difference between respondents with a high (university) and elementary education. In this context, respondents with a higher level of education expressed a more positive attitude towards the impact of tourism development on stopping further emigration.

The results show that a small number of residents are engaged in the tourism sector, bearing in mind that a large majority have never worked in tourism. Despite this fact, the residents believe that the future development of this area is in tourism. Most of them think that the number of tourists is low, but the prevailing opinion is that it has increased after the protected area announcement.

Almost all respondents are aware that they live in a protected area, and a majority of them have a positive attitude toward the protected area. Further, they believe in the positive impact of this fact on their everyday life. Age structure, gender and education have no significant influence on residents' attitudes towards tourism. Generally, residents think that tourism, as a single activity, is not enough for a living. Nevertheless, it could be a factor of emigration reduction and even an incentive factor to settle in this area in the future.

*5.2. Assessment of Vlasina Lake Water Quality*

During the field research and survey conduction, the residents emphasized in informal conversations that they do not use the lake for swimming. They pointed out insecurity as the main reason. The lake shore is unkempt; there are no lifeguards and no ambulance nearby. They also mentioned that the lake water temperature is not so warm in the summer months and that they doubt its good quality because of the lack of sewage and the fact that all wastewater reaches the lake untreated. For this reason, we started analyzing the quality of the lake water, and with the application of water quality indices, we tried to check how justified the residents' fears were.

5.2.1. SWQI Values

In the period 2013–2022, 22 measurements were conducted. The SWQI values range from good (78) to excellent (100). The only exception was one measurement in 2017 when the SWQI value was bad (67) (Table 5). This exception was the consequence of increased values of orthophosphate (0.52 mg/L, which was 18 times higher than the optimal value), BOD (11 mg/L, which was 12 times higher than the optimal value) and suspended solids (48 mg/L, which was 5 times higher than the optimal value), which indicate the organic pollution of lake water.

**Table 5.** SWQI for Vlasina Lake in the period 2013–2022.

| Year | T $q_i \times w_i$ | pH $q_i \times w_i$ | Conductivity $q_i \times w_i$ | OS $q_i \times w_i$ | BOD $q_i \times w_i$ | SS $q_i \times w_i$ | TNO $q_i \times w_i$ | $PO_4^{3-}$ $q_i \times w_i$ | $NH_4$-N $q_i \times w_i$ | *E. coli* $q_i \times w_i$ | SWQI | Description |
|------|------|------|------|------|------|------|------|------|------|------|------|------|
| 2013 | 5 | 9 |   |   | 11 | 7 | 8 | 2 | 6 | 12 | 79 | Good |
| 2014 | 5 | 9 |   |   | 12 | 7 | 8 | 4 | 6 | 12 | 83 | Good |
| 2015 | 5 | 8 |   |   | 13 | 7 | 8 | 4 | 5 | 12 | 82 | Good |
| 2016 | 5 | 8 | 6 | 17 | 10 | 1 | 7 | 7 | 12 | 11 | 84 | Very Good |
| 2017 | 5 | 8 | 6 | 16 | 1 | 2 | 7 | 0 | 12 | 10 | 67 | Bad |
| 2018 | 5 | 9 | 6 | 13 | 13 | 7 | 7 | 7 | 12 | 12 | 91 | Excellent |
| 2019 | 5 | 9 | 6 | 18 | 15 | 7 | 7 | 5 | 11 |   | 94 | Excellent |
| 2020 | 4 | 9 | 6 | 18 | 14 | 7 | 8 | 7 | 12 |   | 97 | Excellent |
| 2021 | 4 | 8 | 6 | 18 | 15 | 7 | 8 | 8 | 12 |   | 98 | Excellent |
| 2022 | 5 | 9 | 6 | 18 | 15 | 7 | 8 | 8 | 11 |   | 99 | Excellent |

In the last several years (from 2018 to 2022), the SWQI values were excellent, while in the first years of the observed period (from 2013 to 2015), the SWQI values were good. According to these results, there could be a tendency for water quality improvement. These results are partially in line with Stevović et al. [43], where the SWQI of Vlasina Lake was very good in 2015. However, Šmelcerović [41] found increased organic pollution in 2020: the BOD was 2.85 mg/L, more than three times higher than the optimal value. The increased organic pollution could be a consequence of municipal wastewater and the erosion of the lake bank [41].

### 5.2.2. CWQI Values

Applying the CWQI methodology, different values have been obtained. The chart presents a comparison between the SWQI and CWQI values for overall water quality (Figure 3).

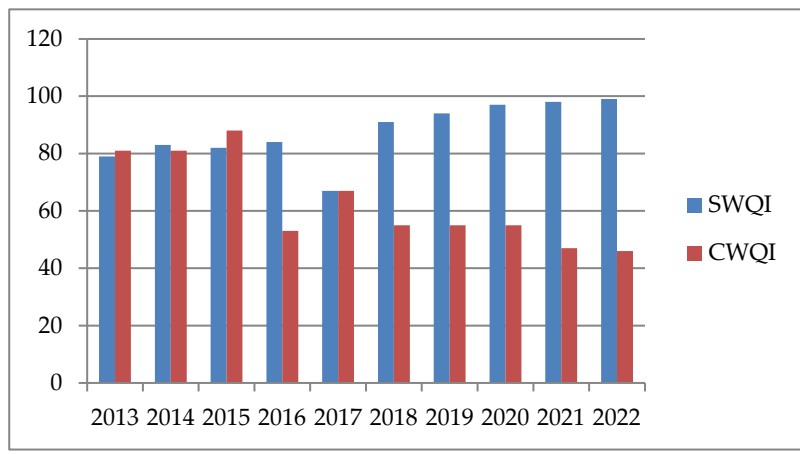

**Figure 3.** SWQI and CWQI for overall water quality in the period 2013–2022.

CWQI values for overall water quality ranged from marginal (46) in 2022 to good (88) in 2015. At the beginning of the observed period (from 2013 to 2015), the SWQI and CWQI values were similar, while in 2017, these two methodologies recorded the same result (67). However, during the other years, the CWQI values were significantly lower than SWQI, especially during the last two years (even more than twice), when the CWQI values for overall water quality were 47 (2021) and 46 (2022). This could be explained by the missing monitoring of the metal concentration (such as As, Cu and Cr) from 2013 to 2015. Taking into account that the variables with the highest normalized sum of excursions (nse) were Cu (from 2018 to 2022) and Cr (2017 and 2016), these parameters were the cause of the lower CWQI values. The variables with the highest nse were Fe during 2013 and 2014 and DO during 2015.

Considering CWQI values for different purposes (Figure 4), the results show the lowest CWQI values for aquatic life, which ranged from poor (37) in 2016 and 2022 to fair (73) in 2015; for drinking water from good (81) in 2022 to excellent (100) in 2015, 2016, 2019 and 2020, while for irrigation the values ranged from good (79) in 2022 to excellent (100) from 2013 to 2015 and 2018. The lowest CWQI values for overall water quality and also various purposes (aquatic life, irrigation and drinking water) were calculated in 2022. Some future investigations should show if this year is the beginning of the water quality decline or just an exception. The highest CWQI values were calculated for livestock and recreation. Excellent CWQI values (100) were recorded for livestock and recreation in all years. However, the CWQI for recreation is based on only one parameter (pH), which is the limitation of CWQI methodology. Investigations of water quality for recreational use in various freshwaters around the world, besides pH values, included more variables such as *E. coli* in the Mngeni River and its tributaries in South Africa [110]; cyanobacteria and turbidity in Lake Pampulha and other Brazilian freshwaters [111,112]; and also detergents, $NO_3^-$, chemical oxygen demand (COD), $PO_4^{3-}$, total coliforms, fecal coliforms and Enterococci in the Portero de los Funes River in Argentina [113]; electrical conductivity (EC), total dissolved solids (TDS), DO, temperature, total hardness (TH) and $Cl^-$ in Colina Lake in Mexico [114].

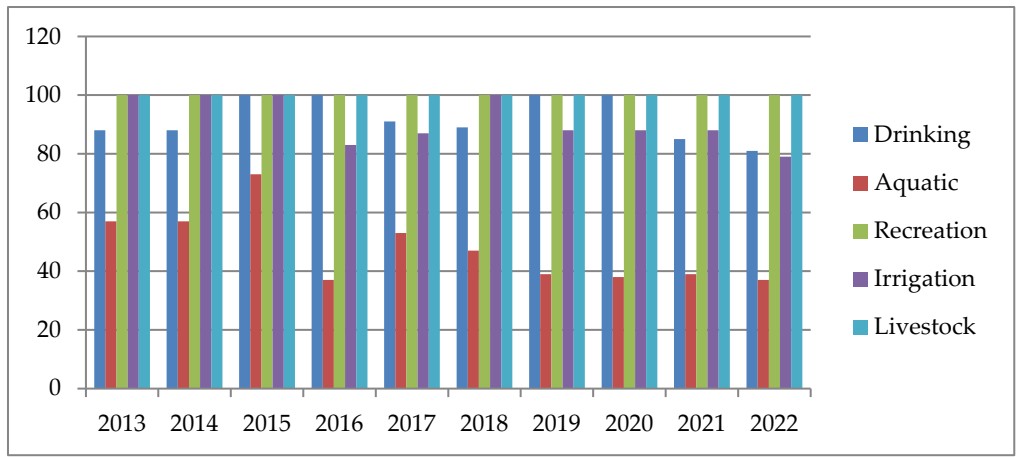

**Figure 4.** CWQI values for different purposes in the period 2013–2022.

5.2.3. WPI Values

The calculated WPI values based on the surface water samples near the dam were in the 0.470–0.815 range, corresponding to pure water. The only exception in the observed period is the WPI value of 1.461 in 2017, corresponding to moderately polluted water. However, this result should be taken with reserve because only one measurement of the parameters was performed that year, unlike other years. The analysis of the WPI values over the observed ten-year period did not reveal any distinct trend, and the obtained results show a low degree of water pollution in Vlasina Lake (Table 6).

Analyzing the values of the parameters used for WPI calculation, it can be concluded that out of 24 parameters, 12 (pH, $NO_2^-$, $NO_3^-$, TN, TP, $Cl^-$, $SO_4^{2-}$, Cu, Zn, Cr, B and dry residue) always belonged to the II/III class, the prescribed class for reservoirs. The values of five analyzed parameters (SS, BOD, $COD_{MnO4}$, TOC and Fe) always belonged to the prescribed threshold values for parameters for the given classes in Serbia that are established at the national level by the several rulebooks and regulations mentioned above. The values of five analyzed parameters (SS, BOD, $COD_{MnO4}$, TOC and Fe) mainly belonged to the prescribed maximum values with minor occasional deviations. The other seven parameters (DO, OS, $NH_4$-N, $PO_4^{3-}$, Mn, As and CB) deviated more or less from the threshold values, affecting the WPI values.

**Table 6.** WPI for Vlasina Lake in the period 2013–2022.

| Year | WPI | Description |
| --- | --- | --- |
| 2013 | 0.815 | Pure |
| 2014 | 0.805 | Pure |
| 2015 | 0.812 | Pure |
| 2016 | 0.601 | Pure |
| 2017 | 1.461 | Moderately polluted |
| 2018 | 0.576 | Pure |
| 2019 | 0.490 | Pure |
| 2020 | 0.470 | Pure |
| 2021 | 0.523 | Pure |
| 2022 | 0.702 | Pure |

According to the analysis, it can be concluded that the indicators of organic pollution ($NH_4$-N, $PO_4^{3-}$ and CB) had a significant impact on WPI values. Ammonium ($NH_4$-N) and orthophosphate ($PO_4^{3-}$) are very important for the formation of nutrient loads. Their increase can cause changes in eutrophication and oxygen depletion. The measured DO and OS values are slightly above permissible limits in this case, but they can indicate the decay of organic matter in the lake. This can cause an increase in the level of aerobic bacteria, which consume oxygen and create less favorable conditions. In addition, DO can indicate pollution by organic matter and the level of water self-purification. However, the measured DO and OS values are slightly above permissible limits in this case. Occasional high values of CB may indicate potential health risks to swimmers.

The oscillations in water quality are primarily related to the issue of wastewater treatment. All settlements of the Vlasina LEF are without adequate sewage infrastructure. They use septic tanks, which are often inadequate. It is important to highlight that there are no public water supply systems in the settlements of the Vlasina LEF, except for part of the population in the Vlasina Rid [84]. This causes problems related to the irregularity of water supply, losses and water quality. An additional challenge for the environment is the inadequate disposal, removal and treatment of solid waste, which results in the formation of illegal landfills [115]. The violation of regulations in their construction and irregular emptying sometimes result in wastewater spillage and groundwater bacteriological contamination. Therefore, it was assumed that the water quality of Vlasina Lake deteriorates over time due to anthropogenic pressures (H3). However, the obtained parameters indicate that this hypothesis is not confirmed. Water quality has not deteriorated over time. In general, the water quality indices values indicate that Vlasina Lake has clear waters suitable for various uses, including tourism and recreation, to a certain extent.

### 5.3. Tourism and Perspectives for Its Further Development in Vlasina LEF

The Vlasina LEF is located mainly in the territory of the Surdulica municipality and a small part of the territory of the Crna Trava municipality, which belongs to the economically highly undeveloped local self-government units in Serbia [93]. In addition, this is a sparsely populated area, with many old and impoverished inhabitants. According to data from the 2022 Census, only 578 residents live in the eight LEF settlements, which is 390 less than in 2011 [102,116]. Due to negative natural growth and emigration, there has been a continuous decline in the number of inhabitants (since 1961) and households (since 1981). On the other hand, there was an increase in the housing stock by about 14.9% (breakup of multigenerational families' construction of facilities for tourist purposes) [117,118]. The age structure in the Vlasina LEF settlements is unfavorable since depopulation and emigration started decades earlier. The population's average age in those settlements was 51.18 years in 2002, 55.05 in 2011, and 56.81 in 2022. In Vlasina Rid, the only settlement with tourist traffic, the average age of the inhabitants is 54.23 [102,116,119]. In addition, initial capital is necessary for tourism, which the residents, primarily unemployed and poor, need to

have. The unfavorable economic structure evidences this because the unemployment rate in Vlasina LEF was 50.6% in 2011 [91].

The Vlasina LEF is protected in order to preserve significant geological and geomorphological features, rich and diverse vegetation, ornithofauna, entomofauna, peat islands and attractive landscapes. There is a three-level protection regime, which must be considered when planning tourism development in this area. For areas in first-degree protection (the islands of Dugi Del and Stratoria, 0.07% surface of the protected area), strict protection is foreseen (controlled visits for scientific research and education). In the areas of second-degree protection (the lake, surrounding mountains, river valleys and streams, 32.90% surface of the protected area), hunting and fishing for scientific and sports-recreational purposes, small objects of eco and rural tourism, rowing or electric boats and the construction of facilities for raising domestic animals and game are allowed. However, it is important to emphasize that constructing the new superstructure is conditioned by constructing a sewerage ring around the lake. Until then, only recreation is foreseen in this area [84]. The areas of the third-level protection (67.03% surface of the protected area) are provided with proactive protection, which allows the limited and selective use of natural resources (collecting forest fruits and medicinal herbs, hunting and fishing are permitted in a certain period of the year), performing of traditional activities, arrangement and revitalization of existing rural buildings and objects of cultural and historical heritage, and the construction of tourist infrastructure and superstructures of smaller size in a conventional style [82].

The Spatial Plan of the Special Purpose Area of the Vlasina LEF singles out Vlasina Lake and the surrounding settlements of Vlasina Rid, Vlasina Okruglica and Vlasina Stojkovićeva as a central tourist zone. However, at the same time, it is also an area of endangered environment. Therefore, any new buildings are conditioned by constructing a sewerage ring around the lake, as mentioned above [84].

Plans for the construction of new tourist resorts and the enrichment of the tourist offer in this area have existed for a long time. A few years ago, the Master Plan for the tourism development of Vlasina Lake was adopted [120]. This plan proposed the formation of five zones in three peripheral settlements, with newly built amenities, in total: 16 hotels, 182 villas, 380 condominiums, 15 townhouses and 106 apartments, including numerous shops, restaurants, bars, boutiques, wellness centers and golf clubs. However, it was not realized.

In 2016, the two largest hotels in the Vlasina LEP ("Promaja" and "Vlasina" along with the camp) were privatized; but in 2019, the investor abandoned the entire project. In mid-2023, the Minister for Promotion of the Development of Underdeveloped Municipalities stated the intentions of a foreign investment group to invest several hundred million euros in this area. They plan to build a hotel, golf course, ski center and marina on the lake, with an obligation to preserve the protected area. The announced investments are presented as particularly significant for the employment of the local population, as it is foreseen to open 10,000 jobs [121]. However, all of these announcements should be taken with caution, due to all past unpleasant experiences.

In addition, the state is finally interested in investing in the infrastructural development of the Vlasina LEF. The sewage system is in the design phase, and its construction is planned to start in the summer of 2024 [122]. Also, in 2021, an international company, in cooperation with the municipality of Surdulica and the tourism organization of the municipality of Surdulica, supported by the UNPD, launched a project entitled "Vlasina—Pure Love". Firstly, workshops for the education of restaurateurs from the area of Vlasina were held, and in 2023, landscape hiking trails with a length of 47 km were renewed, and more than 80 information boards about the wildlife that lives here were installed [123]. This is a good example of successful private and local government cooperation.

According to the Spatial Plan of the Special Purpose Area [84], tourism and recreation are designated as the leading economic branch of sustainable development of the Vlasina LEF, complementary to agriculture. The local population is seen as the primary bearer of tourism development. They should provide potential guests with accommodation, food

(organic), the production and placement of souvenirs and the presentation of the cultural and folk heritage of their region. They are also the most qualified to be tourist guides in these protected natural assets, with specific prior training. This type of engagement would reduce unemployment and increase living standards, which would all positively affect the overall economic development in this area [91].

It is important to emphasize that the tourist-geographic location of Vlasina Lake is not the most favorable. It is far from the mainstream of tourists passing through the Pan-European Corridor X. The network of municipal roads is underdeveloped and in poor condition. Also, there is the abovementioned communal facilities issue, as well as the unsatisfactory condition of social infrastructure (ambulance station, pharmacy, school, culture center, library, post office, sports fields, etc.). Consequently, at this moment, agriculture, forestry, hunting, fishing and tourism (to a certain extent) are dominant activities in the researched area.

Currently, the Vlasina LEF is one of the tourist places with a low spring–summer seasonality, mainly on the local level [124]. The seasonality peak is in July and August, and it is related to excursion tourism, event tourism, and, partly, transit tourism. Due to heavy rains and high winds, spring is unfavorable for tourist activities [92]. Also, the New Year's period can be considered a seasonal peak because the catering facilities on Vlasina are most often filled with tourists from Bulgaria.

Having in mind the mentioned advantages and disadvantages of the Vlasina LEF, the last hypothesis was established. On the basis of objective indicators, as well as the residents' attitudes, tourism activity will contribute to the development of this area in the future. The data of the conducted survey show that more than 70% of the respondents are of this opinion. This fully confirms Hypothesis 4 (H4). However, it should be noted that solving infrastructure problems and improving the state of the environment is a key prerequisite.

Additionally, based on field research from May 2017 to July 2020, interviews with residents, desk research [92,125–128], spatial plans of the special purpose area of the landscape of exceptional features "Vlasina" [84], as well as data collection on the websites of relevant mountaineers, hunters, fishing and cycling associations, resulted in our opinion that the Vlasina LEF favors the development of the following types of tourism:

Sports-recreational and excursion tourism. Vlasina Lake has a long tradition of hosting athletes during preparation. However, nowadays, there are accommodation and accompanying sports facilities issues. For excursionists, there are organized picnic areas, 47 km of marked trails in the surrounding mountains and the possibility of boating. However, few and neglected sports fields, unorganized beaches, the absence of a footpath along the lake and the absence of a parking lot can be cited as a disadvantage for further development. A big disadvantage is a poor connection by public transport with larger settlements in the vicinity and the settlements within the Vlasina LEF, as well as the absence of marked stops with timetables. A school minibus enables transport of residents and visitors, but it does not operate during the holidays and the summer tourist season. A minibus/bus with extremely few daily departures passes through several settlements and does not stop at Vlasina Rid, the only tourist spot in the LEF.

Rural tourism. The settlement that is most suitable for the development of this type of tourism is Vlasina Rid. Numerous households offer accommodation, mostly uncategorized, but the accompanying services need to be improved. Only one modestly stocked store works in July–August, while the market and the info center are in the appropriate place. A prerequisite for developing rural tourism is constructing a sewer ring around the lake. Particular attention should be paid to solving illegal landfills, full of containers that have not been emptied for weeks, as well as health services because the health clinic works only one day a week for a few hours.

Hunting and fishing tourism. It has a good basis for development. The lake is stocked periodically, and a hatchery has been out of operation for twenty years. The coast is accessible, with the possibility of fishing by boat. Fishing is provided on the lake and its tributaries, which are within the protected area's boundaries along the entire length of

their course. The lake's fish stock consists of 22 fish species, including brown trout, catfish, pike, carp, bream and others [129]. It is possible to buy a daily, multi-day and annual license for recreational fishing. They can be bought at the manager's office in Surdulica and the Information Center for Environmental Protection in Vlasina Rid. There used to be two fishing camps, but one was closed after receiving the status because it was in the second protection zone, while the other was closed a few years ago along with its owner, the Vlasina Hotel. Since it is a protected area, special fishing regimes also apply. Fishing is prohibited in areas of the first degree of protection (islands), then from peat islands and in natural spawning grounds. There used to be two fishing camps. This area has three hunting grounds—Vrla, Vlasina and Valmište. They are located near the Vlasina Lake, so hunters who visit them may also be interested in the tourist offer of Vlasina Rid. However, the road is often impassable during the winter and should be reconstructed.

Manifestation tourism. It represents the main axis of cultural tourism offered by the Vlasina LEF. In the Vlasina region, manifestations of a mostly sporting nature are often held during the summer. They are all held in the Vlasina Rid settlement. The most famous and longest-lasting manifestation is the Vlasina Summer, which has been held since 1985. It has regional significance and includes smaller individual competitions (a pre-competition) for the Trumpeter Assembly in Guca; Vlasinski lonac, a competition in preparing fish soup; Owner's hook, a fishing competition; and a Jeep competition. There are also several regional manifestations, such as a regatta, a catfishing competition, the Climb to Čemernik Mountain, Swimming for the Holy Cross, Assembly of St. Elijah and others.

Congress tourism. It has a decades-long tradition in the Vlasina LEF. Unfortunately, the possibilities of hosting this type of tourist have been drastically reduced by the closure of the Vlasina and Jezero hotels. Reestablishing these capacities would contribute to extending the tourist season in Vlasina Lake, which is particularly important since it is very short here.

## 6. Conclusions

Vlasina LEF is one of only three Ramsar sites in the Central part of Serbia that covers Vlasina Lake and the mountains in the southeastern part of the Republic of Serbia, near the border with Bulgaria. The primary natural value of this protected area is Vlasina Lake, an artificial lake, the highest lake in the Republic of Serbia and the second largest. Due to diverse biodiversity and numerous natural resources, this lake and the surrounding area was declared a Natural Asset of Exceptional Importance and protected as a Protected Area of Category I in 2006. Also, Vlasina LEF was protected as an IPA, IBA, PBA and Emerald area. According to the IUCN categorization, it belongs to Category V—Protected terrestrial/marine areas, and it is a Ramsar site. It is an economically poorly developed and depopulated area with a rich natural values and cultural heritage. The present study provides important new insights into the water quality status of Vlasina Lake, as well as residents' perceptions towards local people's involvement in tourist activity and future tourism development in this protected area.

The first part of this study includes the analysis of a survey conducted among residents in the Vlasina LEP. The obtained results showed that a small number of residents are engaged in tourism. They believe that the future economic development of this area is directly related to tourism activity, but most of them think that dealing exclusively with tourism cannot be a sufficient source of income. Analyzing residents' attitudes according to socio-economic variables (gender, age, education), it was concluded that there are no statistically significant differences in their opinions regarding gender and age, but there are statistically significant differences in their attitudes regarding education. Namely, almost three-quarters of both female and male respondents, as well as both older and young people, believe that tourism will contribute to the economic development of this area. In addition, the largest number of respondents with an elementary school education believe that tourism will contribute to the economic development in this region, and only two-thirds of respondents with a university education share their opinion. One of the limitations of this study is that the number of surveyed residents is not evenly distributed

among settlements. Most of the respondents live in three settlements of the Vlasina LEF (Vlasina Okruglica, Vlasina Rid and Božica).

As the central part of this protected area is Vlasina Lake, in this study water quality was assessed in the function of the sustainable tourism development by using SWQI, CWQI and WPI methodology. According to the SWQI, it was determined that the water quality was from good to excellent, indicating a tendency for water quality improvement. The CWQI values for overall water quality ranged from marginal to good in the observed period and excellent for livestock and recreation in all measurements. However, CWQI for recreation is based on only one parameter (pH), which is the limitation of CWQI methodology, so these results should be taken with reserve. Based on WPI values, Vlasina Lake has clean water, suitable for tourism and recreation. Despite this limitation of the WQIs, they are good quantitative tools to provide a general overview of water quality to the public, water managers and decision makers. The limitation of this study is the small number of measurements and parameters in the observed period. Therefore, continuous monitoring of the water quality parameters of Vlasina Lake should be established to obtain more complete and relevant results and recommend eventual measures in the case of water pollution.

The tourist value of Vlasina Lake is primarily reflected in its recreational function and the landscape's beauty. With additional development and the maintenance of infrastructure and superstructure (water supply and sewerage, roads, social standard facilities, recreational and accommodation facilities), as well as adequate tourism valorization, this hydrological object can become a more attractive destination for sustainable tourism and the backbone of the area's development.

Regarding future sustainable tourism development in the Vlasina LEF, the authors are of the opinion that it is not necessary for new accommodation capacity building, considering that existing inactive hotels and resorts can be renovated and reactivated. This would include the reconstruction of the hotels "Vlasina" and "Jezero", as well as the "Vlasina" camp (now an ecological black spot), which would contribute to the extension of the tourist season in Vlasina, increasing of the number of tourists and number of tourist nights and reactivation of congress tourism. In order to complete the tourist offer, it would also be of great importance to construct supporting grounds for sports and recreation. In addition, the existing hatchery, which has been out of operation for twenty years, should be reactivated, considering that the lake is popular among sport fishermen and periodically stocked. Since the area of Vlasina is particularly favorable for the adventure tourism development, it is necessary to improve the offer of appropriate services. This would include a health center with 24 h service and mountain rescue service. Also, it would be very useful to provide stations for the rent and service of bicycles, and shops for hunting and fishing equipment. As manifestations can attract more tourists and represent the main potential for cultural tourism development, it would be appropriate to return the venue of the most established manifestation, "Vlasina Summer" in Vlasina Rid. That was one of the suggestions of all interviewed residents of the Vlasina LEF, together with the proposal to return the seasonal market to the old location. In the end, it is especially important to highlight that it is necessary to open additional tourist info centers in the most visited parts of the protected area, which would provide information to tourists and offer various excursions and programs. Although the Information Center for Environmental Protection is a modern facility, its location is not the most adequate. It is a kilometer away from the main road, and 2.5 km from the tourist center.

Despite the plans for future tourism development, it is also important to emphasize that the most important task is the continuous environmental protection of the Vlasina LEF. This can be supported by cross-border cooperation with Bulgaria, which, as an EU member, has access to many development funds, especially in the segment of environmental protection. In addition, it is also important to include more local communities in this process by supporting its proposals and projects. One of the good examples is the local association "Čuvari Vlasine", founded in 2020, which cooperates with the local self-government,

organizes numerous volunteer actions, manifestations, various educational programs and realizes projects.

The plans for further research include a survey among visitors to analyze their perceptions of tourism activity in this area, perspectives for tourism development, their suggestions for improving the tourist offer and their attitudes towards the environment state and the main environmental issues. In addition, a significant contribution will give qualitative analysis (interviews) for additional clarification of the relations between water indicators and residents' attitudes, as well as residents' attitudes toward tourism activities and its future development. The acquired knowledge will provide guidelines to public policy makers (state bodies, public companies, etc.) for formulating adequate program solutions in the field. Finally, solving infrastructural, social, environmental, demographic and other problems would lead to improving the quality of life of the local community.

**Author Contributions:** Conceptualization: A.M.P.; Methodology: T.J.G., A.M.P. and D.J.; Software: T.J.G., D.J. and A.M.P.; Validation: S.D.; Formal Analysis: T.J.G., D.J. and A.M.P.; Field Research: A.M.P., T.J.G., S.D. and D.J.; Resources: T.J.G. and S.D.; Data Curation: T.J.G. and S.D.; Writing—Original Draft Preparation: A.M.P., T.J.G., S.D. and D.J.; Writing—Review and Editing: A.M.P., T.J.G., S.D. and D.J.; Visualization: T.J.G. and A.M.P.; Supervision: A.M.P., D.J. and S.D. All authors have read and agreed to the published version of the manuscript.

**Funding:** This study was supported by the Ministry of Science, Technological Development and Innovation, Republic of Serbia (Contract No. 451-03-47/2023-01/200172).

**Institutional Review Board Statement:** All subjects gave their informed consent for inclusion before they participated in this study. This study was conducted in accordance with the Declaration of Helsinki, and the protocol was approved by the Ethics Committee of the Geographical Institute "Jovan Cvijić" SASA (114/1).

**Informed Consent Statement:** Informed consent was obtained from all subjects involved in the study.

**Data Availability Statement:** Collected data are gathered in the SPSS data basis matrix.

**Acknowledgments:** We are grateful to the members of the local community of the Vlasina Landscape of Exceptional Features for participation in the survey, and to the Joint Stock Company "Elektroprivreda Srbije", especially Miodrag Marković from the Vlasina hydropower plant in Surdulica (branch of Đerdap hydropower plant) for providing us with data on water quality parameters. Also, we are grateful to Jovana Brankov, for constructive suggestions for additional literature selection.

**Conflicts of Interest:** The authors declare no conflict of interest.

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
