# Peer review of "Sustainable Tourism Development and Ramsar Sites in Serbia: Exploring Residents’ Attitudes and Water Quality Assessment in the Vlasina Protected Area"

_sustainability, doi:10.3390/su152115391_

Round 1
Reviewer 1 Report
The research paper assess the potential for sustainable development of tourism on Vlasina Lake by conducting various methods of water quality assessment and also taking into consideration the perceptions of the local people through a survey. The authors indicated that the residents of Vlasina LEF are "primarily unemployed and poor", and therefore the initial capital needed for tourism is absent among the locals. On the other hand it is also indicated that there are plans to construct new tourist complexes for active tourism development. The authors need to explain in a little more details about who plans this, where will the money for tourism infrastructure development come from and how will the locals, who are unable to provide capital, benefit in the process.
Author Response
Dear Reviewer,
I would like to thank you for carefully reading the manuscript, constructive suggestions and helpful comments that helped us to improve the quality of the manuscript. Enclosed you will find a Cover letter with our answers to your comments.
Kind regards,
Authors

Reviewer 2 Report
The article investigates sustainable tourism development in Serbia's Vlasina Protected Area, specifically examining residents' attitudes and conducting water quality assessments. The study sheds light on the perceptions of local residents and the ecological aspects of tourism in a Ramsar site.
Here are my suggestions for enhancing the paper:
Introduction. While the introduction references a range of international studies and initiatives related to lake conservation, it would be beneficial to emphasize the local context and relevance of the study, particularly for readers less familiar with Serbia. The introduction could benefit from a clear and concise statement of the research objectives to provide a roadmap for the reader.
The article lacks a Literature Review section. The provided list of references appears to be extensive and relevant to the topic of lake ecosystems and their management, with a particular focus on sustainable development. However, the list includes references from as early as 2003 to as recent as 2023. While this can provide a comprehensive overview of the literature, it's essential to consider the currency of the information, especially in rapidly evolving fields such as environmental science. Some references might be outdated in terms of the latest research findings and policies. I suggest incorporating the following references that provide a combination of sustainability, tourism, and regional analysis, which can contribute to the discussion in the paper on sustainable tourism development and water quality assessment in the Vlasina Protected Area in Serbia.
Santos, E., Carvalho, M., & Martins, S. (2023). Sustainable Water Management: Understanding the Socioeconomic and Cultural Dimensions. Sustainability, 15(17), 13074.
This reference provides insights into the socio-economic and cultural aspects of water management, which could be relevant when discussing water quality assessment in the context of sustainable tourism.
Santos, E., Lisboa, I., Moreira, J., & Ribeiro, N. (2021). The Productivity Puzzle in Cultural Tourism at Regional Level (pp. 371-383). Springer.
This reference provides valuable insights into the broader context of sustainable tourism and regional-level analysis, which could be applicable when discussing the development of sustainable tourism in the Ramsar sites in Serbia.
Study área. The section contains an abundance of geographical and historical information about Vlasina LEF. While this information is undoubtedly important, it could be presented in a more concise manner. To maintain reader engagement, it's essential to strike a balance between providing necessary context and avoiding information overload. While it's important to provide background information about the area, it's crucial to tie this information back to the main focus of the study – sustainable tourism development and water quality assessment. The section could emphasize the relevance of the area's features to the study's objectives. Also, the section could benefit from clearer subheadings or formatting to break down the information into easily digestible chunks. This would help the reader navigate through the details more efficiently. The section briefly mentions the economic development of the area but doesn't elaborate on the potential impact of tourism on the region's economy. Providing some insight into how sustainable tourism might contribute to economic development could be beneficial. Moreover, the protection status of the area is mentioned, but it could be explained in more detail, especially regarding how this designation might impact tourism and environmental conservation efforts. The last sentence mentions the need to balance economic development and environmental protection. While this is a crucial point, it could be expanded upon to emphasize its significance within the context of the study.
Methodology. While it's essential to justify the survey's methodology, the lengthy list of references and descriptions of previous studies might be more suitable for the literature review section. The section could be streamlined for a more focused presentation of the survey methodology. The section mentions that the survey was conducted on a sample of 81 respondents, representing approximately 9.6% of the adult population in the area. While it's acknowledged that this might be the best possible sample given the population size, it's important to acknowledge the limitations of the sample size and discuss how this might impact the generalizability of the survey findings. It might be helpful to provide a brief summary or rationale for why multiple indices were chosen and how they complement each other in assessing water quality. This can help the reader understand the significance of using multiple indices. It's essential to discuss any potential limitations of using historical data and how the data collection frequency might affect the accuracy and reliability of the water quality assessment. In the water quality classification table, there is some inconsistency in terminology between the SWQI, CWQI, and WPI columns. For clarity, ensure consistent terminology and descriptions across all columns. While the table provides classification values for water quality, it would be beneficial to interpret what these classifications mean for the study area. For example, how do these classifications relate to the potential for sustainable tourism development or other environmental considerations?
Results. The section lacks a concise summary of the main findings, which could provide readers with a clear roadmap of what to expect. The use of subheadings could significantly enhance the organization of the section, making it easier for readers to navigate through the results. Although statistical tests (e.g., chi-square tests) are mentioned, specific findings and significance levels are missing, making it challenging to understand the relationships between variables. The discussion of findings should include context and explanations for residents' attitudes, along with qualitative insights or quotes from respondents to enrich the discussion. The presentation of water quality assessment results needs more clarity, with a focus on highlighting specific findings related to water quality indices and trends over time. The implications of water quality findings for tourism development and environmental conservation could be explored more comprehensively, considering how water quality impacts these aspects and suggesting potential actions or policies. The section's organization and flow could be improved, with concise headings and subheadings to guide readers, and more concise, straightforward writing to enhance readability. Redundancy should be avoided, particularly in repeating information about water quality classifications. Ensure consistent terminology and language use throughout the paper while checking for typographical errors or language inconsistencies.
Conclusion. A concise summary of key findings regarding water quality, socio-economic variables, and tourism potential could enhance clarity. Conclusions should ideally include recommendations or implications based on the research findings. Authors could suggest potential actions, policies, or strategies that should be considered to address water quality concerns, promote sustainable tourism, or balance environmental protection and tourism development. The section should also establish a clear link with the introduction, explaining how the study's findings contribute to the broader research context or address the research questions or hypotheses set out at the beginning. Make sure that the conclusions are presented in a clear and concise manner. Avoid unnecessary repetition of information and focus on summarizing the most significant outcomes of the study. Briefly mention potential avenues for future research.
The text should maintain clarity and conciseness consistently across the paper, while also avoiding repetition and typographical mistakes.
Author Response

(The authors gave the same response as above.)

Reviewer 3 Report
The paper contemplated relevant topics. However, this study should be improved in terms of:
1 - The main question addressed by the research - a better relationship between the aspects analyzed - water indicators and residents’ attitudes – should be established. Another option can be focus in one of the aspects (water indicators or residents’ attitudes);
2 - Methodology: i) It would be more useful the implementation of qualitative analyses, such as interviews, to a deeper and wider investigation of “Residents’ attitudes toward tourism activities and its future development”. In this sense, it is important to highlight that, according to this study, “a small number of residents are engaged in tourism”. The application of qualitative analyses can also facilitate the establishment of clearer and more direct relationships between water indicators and residents’ attitudes; ii) It is important to present the survey questions as it was presented to the respondents.
3. Better understanding of the context analyzed through the presentation of the results of the studies instead of a list of them: “Sustainable management of Ramsar sites has been 68 studied in Songor Lagoon Ramsar site in Ghana [17] and Boondall Wetlands Reserve in 69 Australia [18]. Perceptions of ecotourism potential from residents in Ramsar sites have 70 been studied in the Lake Natron Ramsar site [19] and Kilombero Valley Ramsar site in 71 Tanzania [20], as well as in Moulouya Estuary in Morocco [21]. In addition, perceptions 72 and attitudes of residents toward wetland conservation have been analyzed in Evros and 73 Axios Ramsar sites in Greece [22], Xuan Thuy National Park Ramsar site in Vietnam [23], 74 Bumdeling in Bhutan [24], Mare Aux Cochons in Seychelles [25], Carska Bara Special 75 Nature Reserve [26], Landscape of exceptional features "Vlasina" [27], as well as in Gornje 76 Podunavlje and Koviljsko-Petrovaradinski rit [28] in Serbia. Residents' perceptions and 77 preferences towards natural resources have been studied in U Minh Thoung National 78 Park in Vietnam [29]; residents' perception of the impacts of drought on wetland and 79 household benefits in Driefontein Grasslands in Zimbabwe [30] and residents willingness 80 to pay for restoration in Ashtamudi Lake in India [31]. A study of anthropogenic impacts 81 on Ramsar sites has been performed in Koshi Tappu Wetland in Nepal [32]. The connec- 82 tion between tourism and nature protection has been explored in various Ramsar sites in 83 Serbia: Gornje Podunavlje, Slano Kopovo, Zasavica, Labudovo Okno [33], as well as in 84 Ramsar sites of other countries, such as Lake Butrint in Albania [34], Johor Ramsar sites 85 in Malaysia [35] and Deepar Beel Wetlands in India [36]. Socio-economic potential of the 86 Ramsar site has been examined in Carska Bara Special Nature Reserve [37]. 87 Considering that lakes often have recreational role and possible pollution can affect 88 human health, various lakes have been studied for their appropriateness for recreational 89 purposes: Colina Lake in Mexico [38], Shahu Lake in China [39], Pushkar Lake in India 90 [40], Ahor Lake in Ghana [41], Lake Ostrovąs [42] and Lake EÅ‚k [43] in Poland, artificial 91 lake Kisköre Reservoir (Lake Tisza) in Hungary [44]. Natural conditions for the devel- 92 opment of lake tourism have been analyzed in Poland [45]. On the other hand, the im- 93 pacts of tourism activities on water pollution have been investigated in the West Lake 94 Basin in China [46] and dune lakes on Fraser Island in Australia [47]. Lake tourism has 95 been the subject of various studies around the world. It has also been presented at lake 96 tourism conferences, which were held in: Savonlinna in Finland (2003), Thousand Islands. Lakes in Hangzhou in China (2005), Gyöngyös in Hungary (2007) and Thunder Bay in 98 Canada (2009) [48]. Residents' attitudes towards ecotourism and conservation have been 99 studied in Five Blues Lake National Park in Belize [49], residents' perceptions of the en- 100 vironmental impacts of tourism in the Lake Bosomtwe in Ghana [50], tourist perceptions 101 have been analyzed in the tourist area of the Lake Toba in Indonesia [51], both residents 102 and tourist perceptions in Lake Balaton in Hungary [52] and Lake Mjøsa in Norway [53]. 103 In contrast, cultural tourism has been examined in Crater Lake in the USA [54]. The im- 104 pacts of second-home tourism on the environment have been discussed in the Lake Dis- 105 trict in Finland [55]. Tourism development and sustainability have been studied in 106 Windermere's Lake in the UK [56], Lake Eyre and Lake Eildon in Australia, Lake Taupo 107 and Lake Wanaka in New Zealand [57] and Lake Salda and its environment in Turkey 108 [58]. A study of lake tourism types has been performed in four lakes in Hungary: Lake 109 Balaton, Lake Tisza, Lake Velence and Lake Fertö [59].”
4. The indication of the studies related to each quotation. Examples: i) Unreferenced statements -“Freshwater ecosystems, including lakes and wetlands, are considered essential life 28 assets. However, these ecosystems occupy relatively limited parts of the Earth.”; Unlike natural 111 lakes that are smaller in size and importance, reservoirs are of great importance and most 112 often have multifunctional purposes. Their purpose is primarily to supply water to the 113 population and industry, energy, irrigation, flood protection, and traffic. They are used to 114 a lesser extent for fishing needs and, more recently, for sustainable tourism development. 115; ii) Missing reference information - “According to the World Lake Vision Committee (Year?), lakes are an essential component of global water resources. The lake's role is fundamental in the continuing cycle of evaporation, precipitation and water flow on and under the land surface. Lakes are storage bodies for large water quantities, sources of food and recreational pleasure for humans, as well as habitats for various aquatic organisms. During flood events, lakes have the ability to mitigate flood waves and, in that way, protect lives and properties.” + “International network Living Lakes aims to improve the protection, restoration and rehabilitation of lakes, wetlands and other freshwater bodies and their catchment areas (reference?).”
5. It can be considered “more academic” to avoid phrases such as “as far as we know”. Please, base your affirmation, related to the level of innovation of this study, on quotations.
Some publication suggestions:
Arbogast, D., Butler, P., Faulkes, E., Eades, D., Deng, J., Maumbe, K. and Smaldone, D. (2020), Using social design to visualize outcomes of sustainable tourism planning: a multiphase, transdisciplinary approach, International Journal of Contemporary Hospitality Management, Vol. 32 No. 4, pp. 1413-1448. https://doi.org/10.1108/IJCHM-02-2019-0140
Hanafiah, M. H., Jamaluddin, M. R., & Kunjuraman, V. (2021). Qualitative assessment of stakeholders and visitors perceptions towards coastal tourism development at Teluk kemang, port dickson, Malaysia, Journal of Outdoor Recreation and Tourism, 35, https://doi.org/10.1016/j.jort.2021.100389
Hsieh, C.-M., Tsai, B.-K., & Chen, H.-S. (2017). Residents’ Attitude toward Aboriginal Cultural Tourism Development: An Integration of Two Theories. Sustainability, 9(6), 903. MDPI AG. Retrieved from http://dx.doi.org/10.3390/su9060903
Sanja Obradović & Vladimir Stojanović (2022) Measuring residents’ attitude toward sustainable tourism development: a case study of the Gradac River gorge, Valjevo (Serbia), Tourism Recreation Research, 47:5-6, 499-511, DOI: 10.1080/02508281.2020.1870073
Sdrali, D., Goussia-Rizou, M. & Kiourtidou, P. Residents’ perception of tourism development as a vital step for participatory tourism plan: a research in a Greek protected area. Environ Dev Sustain 17, 923–939 (2015). https://doi.org/10.1007/s10668-014-9573-2
Minor editing of English language required
Author Response

(The authors gave the same response as above.)

Round 2
Reviewer 3 Report
The main question addressed by the research - a better relationship between the aspects analyzed - water indicators and residents’ attitudes – could be established. Another option can be focus in one of the aspects (water indicators or residents’ attitudes).
It would be more useful the implementation of qualitative analyses, such as interviews, to a deeper and wider investigation of “Residents’ attitudes toward tourism activities and its future development”. In this sense, it is important to highlight that, according to this study, “a small number of residents are engaged in tourism”. The application of qualitative analyses can also facilitate the establishment of clearer and more direct relationships between water indicators and residents’ attitudes
Regarding the survey, it is important to present the survey questions as it was presented to the respondents instead of the topics.
Minor editing of English language required
Author Response
Dear Reviewer,
we would like to thank you for your comments. Enclosed you will find a Cover letter with our detailed responses. All new changes are highlighted in green.
Kind regard,
Authors

Round 3
Reviewer 3 Report
.
Minor editing of English language required